# Self-rechargeable cardiac pacemaker system with triboelectric nanogenerators

Hanjun Ryu [1,6], Hyun-moon Park[2,6], Moo-Kang Kim[3], Bosung Kim[1], Hyoun Seok Myoung[2], Tae Yun Kim[1], Hong-Joon Yoon [1], Sung Soo Kwak[1], Jihye Kim [1], Tae Ho Hwang[4], Eue-Keun Choi[3] & Sang-Woo Kim [1,5✉]

Self-powered implantable devices have the potential to extend device operation time inside the body and reduce the necessity for high-risk repeated surgery. Without the technological innovation of in vivo energy harvesters driven by biomechanical energy, energy harvesters are insufficient and inconvenient to power titanium-packaged implantable medical devices. Here, we report on a commercial coin battery-sized high-performance inertia-driven tribo-electric nanogenerator (I-TENG) based on body motion and gravity. We demonstrate that the enclosed five-stacked I-TENG converts mechanical energy into electricity at $4.9\,\mu W/cm^3$ (root-mean-square output). In a preclinical test, we show that the device successfully harvests energy using real-time output voltage data monitored via Bluetooth and demonstrate the ability to charge a lithium-ion battery. Furthermore, we successfully integrate a cardiac pacemaker with the I-TENG, and confirm the ventricle pacing and sensing operation mode of the self-rechargeable cardiac pacemaker system. This proof-of-concept device may lead to the development of new self-rechargeable implantable medical devices.

[1] School of Advanced Materials Science and Engineering, Sungkyunkwan University (SKKU), Suwon, Republic of Korea. [2] Research and Development Center, Energy-Mining LTD., Seoul, Republic of Korea. [3] Department of Internal Medicine, Seoul National University Hospital, Seoul, Republic of Korea. [4] SoC Platform Research Center, Korea Electronics Technology Institute (KETI), Seongnam, Republic of Korea. [5] SKKU Advanced Institute of Nanotechnology (SAINT) and Samsung Advanced Institute for Health Sciences & Technology (SAIHST), Sungkyunkwan University (SKKU), Suwon, Republic of Korea. [6] These authors contributed equally: Hanjun Ryu, Hyun-moon Park. ✉email: kimsw1@skku.edu

Body-implantable bioelectronics devices that monitor and modulate abnormalities in patients are highly sophisticated technologies. The long-term operation of such in vivo devices has faced major technological challenges. Currently, high-risk repeat surgeries are required to replace used medical implant devices. The permanent operation of body-implantable bioelectronic devices will reduce both the financial burden and the health risks associated with surgery, including device removal and replacement[1–3]. Though a variety of in vivo energy harvesters that use near-field or mid-field electromagnetics[4–6], thermal gradients[7], and the mechanical movement of organs[8–17] has been proposed, in vivo power generators remain limited and currently lack sufficient power to charge the bioelectronic device batteries. Furthermore, the titanium (Ti) packaging of implantable devices interferes with energy transfer, resulting in low energy conversion efficiency[18]. In addition, energy transfer systems require external energy transmitters that can be inconvenient for the user[6]. Thus, for now, batteries, which inherently have a finite lifespan, are the only possible energy source for implantable medical devices.

The development of electret and electromagnetic microelectromechanical system (MEMS) power generators driven by elastic motion based on a spring suspension is a trailblazing approach, and it is appropriate for standalone medical devices[19]. Unlike other in vivo energy harvesting systems, MEMS power generation systems can be driven by human motion at a few Hz without external energy transmitters using spring suspensions[20]. However, relying on the resonant frequency of spring suspensions does not work well for a body-implantable device, because there is no continual body movements[21]. Also, the energy required by a cardiac pacemaker that has a conventional solid lithium-iodine battery with a 2.3 Ah capacity[22], but low root-mean-square (RMS) output around the hundreds of nW of MEMS power generators, may not be sufficient to charge an energy storage device[21]. Spring suspensions are inefficient in that they constrain device mass, displacement, frequency, space, and vibration direction. Because of the multi-directional nature of body motion, a MEMS power generator shows off-axis actuating, which decreases the output power efficiency and stability of the device. Therefore, the development of in vivo energy harvesting systems or some other approach is required[23–26].

In this paper, we demonstrate a commercial coin battery-sized high-performance inertia-driven in vivo triboelectric nanogenerator (I-TENG) based on body motion and gravity constructed using amine-functionalized poly(vinyl alcohol) (PVA-NH$_2$) and perfluoroalkoxy (PFA) as triboelectric materials. We also successfully operated the in vivo stacked I-TENG in a preclinical test and collected real-time output-voltage data via a Bluetooth low-energy (BLE) information-transmitting system. Furthermore, we successfully demonstrated a self-rechargeable cardiac pacemaker system that recharges its battery using the I-TENG. The synchronous stack structure, which achieves current waveform superposition, can step the peak current value up without additional components so that the five-stacked I-TENG generated 136 $V_{peak}$ and 2 $\mu A_{peak}$/cm$^3$. The maximum volume power density of the I-TENG was 4.9 $\mu W_{RMS}$/cm$^3$ at a load resistance of ~10 MΩ in a laboratory experiment, which is a competitive high output performance[11–17]. We found that the I-TENG was able to harvest z-axis mechanical energy and was not strongly influenced by x- and y-axis motion. Fully encapsulated I-TENGs inserted at different places in a large animal had different normal directions and behaved independently. The device harvested around 144 mW in the preclinical large animal experiment, and the activity time of the stacked I-TENG is less than 20% of the total experiment time due to the behavior of the animal in its cage. Furthermore, the stacked I-TENG charged capacitors even from small movements while an adult mongrel was asleep, and charged lithium (Li)-ion battery with the help of a power management integrated circuit (PMIC) in the preclinical test. Finally, we successfully

integrated a cardiac pacemaker with the I-TENG and demonstrated the VVI mode of the self-rechargeable cardiac pacemaker. Thus, we suggest that the I-TENG is a promising in vivo energy harvester for low-power implantable electronic devices.

## Results

**In vivo inertia-driven TENG**. Gravity, which affects all objects, is one of the major environmental energy sources that affect humans; conversely, humans consume metabolic energy to counteract the effects of gravity[27]. During human locomotion, interactions between energy and gravity create a regular inertia effect within the body, which is normally wasted. In spite of its low energy density, this wasted inertial energy can be a valuable source of energy for TENGs for biomedical implant devices, which consume an extremely small amount of power[28,29]. During walking, the vertical height of the chest periodically changes by about 6 cm (see Fig. 1a)[30,31], which would influence a device inside the body. Figure 1b shows a schematic image of a cylindrical I-TENG with a radius of 1.5 cm and a height of 2.4 mm that is used to harvest vertical body motion. In this device, two different triboelectric layers, PFA and PVA-NH$_2$ layers served as the negative and positive triboelectric materials, respectively (Supplementary Fig. 1 and Supplementary Note 1 discuss the properties of PVA-NH$_2$), and a freestanding copper (Cu) mass was used for the inertial movement. While the body moves upward, the whole I-TENG receives the same upward force (see Fig. 1c(i)). As the body begins to move downward, the I-TENG's packaging, substrate, gold (Au), and PVA-NH$_2$ also receive a downward force, but only the PFA/Cu/PFA (i.e., the freestanding unit) moves upward, and it contacts the top PVA-NH$_2$ triboelectric layer due to inertia (see Fig. 1c(ii)). Then, gravity drags the freestanding unit downward to make contact with the bottom PVA-NH$_2$ triboelectric layer (see Fig. 1c(iii)). This behavior of the freestanding unit will be repeated numerous times in the course of a normal day; thus, inertial energy is converted into electrical energy (Supplementary Note 2 and Supplementary Fig. 2 discuss in detail the working mechanism of the I-TENG). Figure 1d and Supplementary Fig. 3 show a real photographic image of the I-TENG and a commercial coin battery (CR3032). To maximize the charge-transfer efficiency between the PFA and PVA-NH$_2$ in the physically limited conditions of the implantable device, the gap distance between the triboelectric materials was optimized by finite element method simulation (see Supplementary Fig. 4). In addition, surface-treated PFA was used as a triboelectric material to improve the energy conversion efficiency (Supplementary Note 3 and Supplementary Fig. 4 discuss in detail the design of the I-TENG). Figure 1e and Supplementary Fig. 5 show a power management system that was developed for the stacked I-TENG in order to charge a battery. For real-time analysis of the I-TENG driven by biomechanical energy during the in vivo experiment, the BLE-based wireless monitoring system was developed to transmit the real-time output information from the stacked I-TENG to a smartphone (see Supplementary Fig. 6). The entire device was fully encapsulated and packed with a 5 mm-thick coating of a commercial biocompatible polymer (C6-540 Liquid Silicone Rubber, Dow Corning) to prevent cross-contamination between the device and the body.

**Characterization of the I-TENG**. Typically, vertical-mode triboelectric nanogenerators have a high open-circuit voltage and a low short-circuit current because of their high optimum impedance[32]. Although the maximum power of the TENG at optimum impedance is significant, the small current output at the system impedance is a critical limitation[33]. Thus, a connection of multiple I-TENGs in

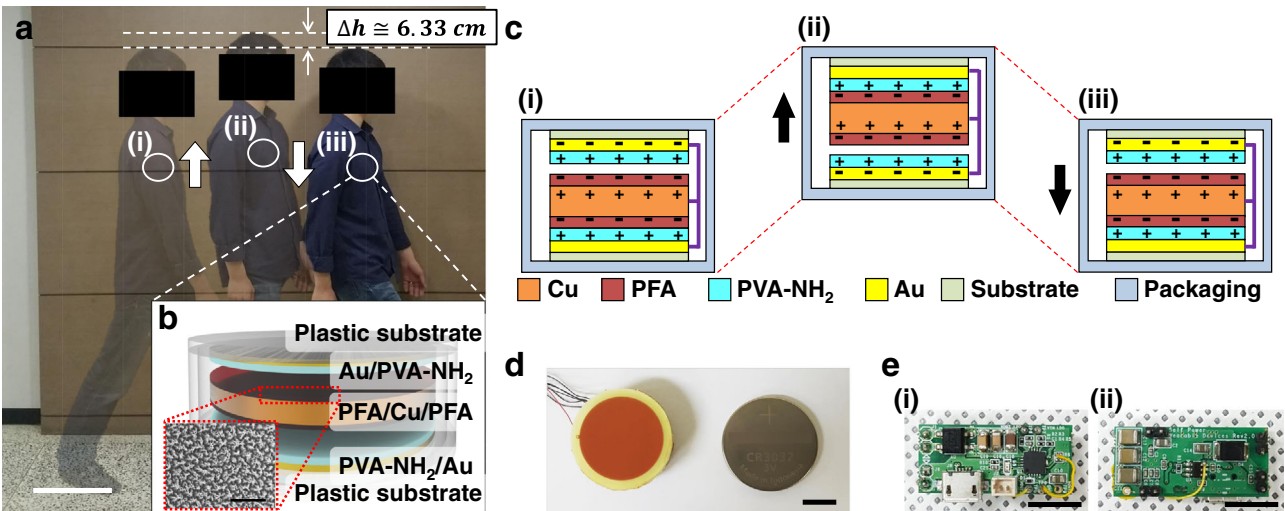

**Fig. 1 Overview of the in vivo triboelectric nanogenerator driven by inertia. a** Photograph of a walker showing a vertical displacement of ~6 cm. scale bar, 30 cm. **b** Schematic image of the I-TENG, and surface SEM image of the surface-modified PFA layer. scale bar, 5 μm. **c** The working mechanism of the I-TENG with the freestanding unit. **d** Photograph of the stacked I-TENG and a commercial coin battery. scale bar, 1 cm. **e** Photographs of the (i) front and (ii) back of the power management system. scale bar, 1 cm.

parallel to increase current output seems to be a promising approach (see Fig. 2a). Figure 2b, c shows the simulation results of asynchronous and synchronous TENGs output current waveforms produced with a simulation program with integrated circuit emphasis (SPICE) simulation. In general, it is difficult to operate different TENGs in the same electrical phase, because to do so, all of the TENGs would have to receive the exact same mechanical energy at the same moment. If five different TENGs operate with five different phases, the total output current is not significantly increased, because of the current waveform offset (see Fig. 2b). This current offset can be prevented with the help of electrical circuits, but additional electronic components, such as rectifiers that rectify each independent current waveform, reduce energy management efficiency[34,35]. However, because the encapsulated design of I-TENGs uses indirect mechanical force to oscillate the freestanding unit, the external mechanical energy applied to the stacked I-TENG is able to synchronize the five I-TENGs in the same electrical phase. Because of the identical current phases, the five waveforms are superimposed and the current is amplified without the need for additional electronic components (see Fig. 2c).

To confirm that the I-TENG performance depends on the number of stacked devices under a 3 Hz shaking condition (see Supplementary Fig. 7), output voltage and current were measured at a load resistance of 10 MΩ. As the number of stacked I-TENGs increased from one to five, the peak voltage increased from 36 to 136 V, and the peak current volume density increased from 0.4 to 2 μA/cm³ (see Fig. 2d, e). The five stacked I-TENGs generated similar voltage under different frequencies (see Supplementary Fig. 8). Figure 2f shows the volume output current and power density of the five-stacked I-TENG as a function of load resistances. The five-stacked I-TENG had a maximum power of 4.9 μW$_{RMS}$/cm³ with ~10 MΩ matched load resistance, and stable output of over 30,000 cycles (see Supplementary Fig. 9). The internal energy conversion efficiency of the I-TENG can be defined as the ratio between the output electric energy and the input kinetic energy of the I-TENG, which is expressed as

$$\text{Internal energy conversion efficiency} = \frac{\text{Output electric energy}}{\text{Input kinetic energy}} \times 100(\%)$$
(1)

The upward input kinetic energy of the I-TENG is derived as

$$\text{Input kinetic energy} = \left[\frac{1}{2}mv^2\right]_{v_{initial}}^{v_{max}}$$
(2)

where $m$ is the weight of the device and $v$ is the velocity of the I-TENG. The output electric energy of the I-TENG is expressed as

$$\text{Output electric energy} = W_{RMS} \times (t_1 - t_0)$$
(3)

where $W_{RMS}$ is the RMS output power of the I-TENG and $t_1 - t_0$ is one output period. The input kinetic energy of the I-TENG is 4.845 mJ, and the output electric energy of the five-stacked I-TENG is 11.43 μJ. Therefore, the internal energy conversion efficiency of the I-TENG is around 0.235% (Supplementary Fig. 10 and Supplementary Note 4 discuss in detail the calculation of the internal energy conversion efficiency).

Based on the laboratory experimental results, we evaluated the power management system and demonstrated a small battery (ML414H, Seiko Instruments Inc., 1 mAh capacity) charging using the five-stacked I-TENGs. First, the output power of the I-TENG device charges parallel-connected low-equivalent series inductance (ESL) capacitors with a capacitance of 60 μF (i.e., first energy storage) after filtering through a low-pass filter (see Fig. 2g). Then, in order to manage the high voltage of the first energy storage, the 1st energy storage discharges to larger energy storage with the help of a PMIC. The larger energy storage that consists of parallel-connected low-equivalent series resistance capacitors with a capacitance of 150 μF (i.e., second energy storage) could store the same amount of energy at most 0.4 times lower voltage (see Fig. 2h). After the second energy storage is charged up to 3.4 V, the stored energy transfers to a Li-ion battery (see Fig. 2i).

We further confirmed the relationship among output performance, displacement, and direction gradient of the five-stacked I-TENG. Human motion creates force not only in the z-axis direction but also in the x- and y-axis directions. As shown in Fig. 3a–g, the net force ($F_{net}$) applied to the freestanding unit is the sum of the human motion-induced inertial force ($F_{Inertia}$) and the force of gravity ($F_{Gravity}$), which is expressed as

$$F_{net} = F_{Inertia} + F_{Gravity} = m\vec{a} + m\vec{g}$$
(4)

where $m$ is the weight of the freestanding unit, $\vec{a}$ is inertial

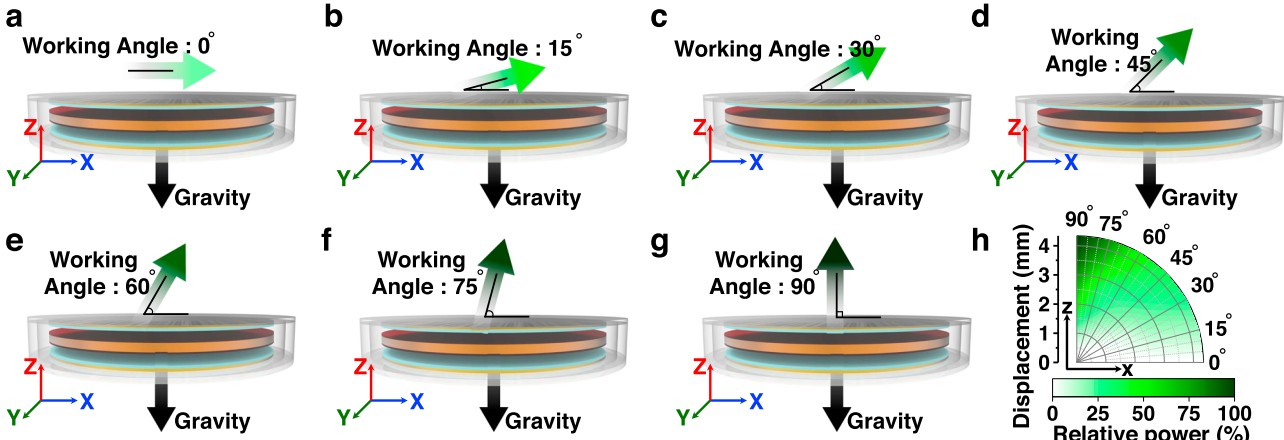

**Fig. 2 The power performance of the I-TENG in the laboratory environment. a** A schematic image of the stacked I-TENG. SPICE simulation results of (**b**) asynchronous TENGs and (**c**) synchronous TENGs. **d** Voltage output performance and (**e**) current output performance of one to five-stacked I-TENGs. **f** Output RMS current and RMS power performance of the five-stacked I-TENG as a function of the external load resistance. Charging behaviors of (**g**) first and (**h**) second energy storages in PMIC and (**i**) battery by the stacked I-TENG.

**Fig. 3 I-TENG power performance according to displacement and working angle. a–g** Demonstration that the I-TENG's displacement direction depends on the working angle. **h** The output voltage distribution of the I-TENG according to displacement and working angle of movement.

acceleration, and $\vec{g}$ is gravity. The converted lateral and vertical components of the three-dimensional $\vec{a}$ and $\vec{g}$ values are expressed as

$$m\vec{a} + m\vec{g} = ma_{Lat.} + ma_{Vert.} + mg_{Vert.} = (ma(t)\cos\theta)\vec{x} + (ma(t)\sin\theta - mg)\vec{z} \quad (5)$$

where $a(t)$ is the time-dependent acceleration of inertia and $\theta$ is the working angle of the device. Because the lateral movement of the freestanding unit could not induce electrostatic induction through an external circuit, we analyzed the relationship between vertical displacement and output voltage. The vertical displacement of the freestanding unit ($d_{ver.}$) and the open-circuit output voltage ($V_{OC}$) are decided by

$$d_{ver.} = \iint_0^T (a(t)\sin\theta - g)dt = \sin\theta \iint_0^T (a(t))dt - \frac{1}{2}gT^2 \quad (6)$$

$$V_{OC} = \frac{\sigma}{\varepsilon_0}d_{ver.} = \frac{\sigma}{\varepsilon_0}\left[\sin\theta \iint_0^T (a(t))dt - \frac{1}{2}gT^2\right] \quad (7)$$

where $T$ is the time duration of human walking, $\sigma$ is the surface triboelectric charge density, and $\varepsilon_0$ is vacuum permittivity. As shown in Fig. 3h, we measured the five-stacked I-TENG at intervals of 15 degrees of the working angle at various accelerations, which resulted in displacement from 0 to 4.3 mm of the I-TENG at diverse working angles. The I-TENG has an almost linear relationship with the vertical displacement of the freestanding unit and the output voltage of the device; x-axis displacement rarely interferes with the device performance; z-axis displacement of more than 4 mm allows the I-TENG to have an operating efficiency of over 80%. Supplementary Fig. 11 shows the energy conversion efficiency of each experimental condition. Thus, a large z-axis displacement or strong acceleration is required to achieve sufficient output voltage. Although body motion has a lower acceleration force than in the experimental conditions, sufficient z-axis displacement of the I-TENGs could easily be obtained because of the long acceleration time, which might be enough to obtain a sustainable energy harvest.

**In vivo evaluation of the stacked I-TENG.** In order to evaluate the potential of the stacked I-TENG driven by biomechanical energy as a power source for implantable medical devices, the packaged devices were implanted in the back of an adult mongrel and their in vivo performance was characterized (see Animal preparation in "Methods"). Real-time output voltage data and charged capacitor voltage data were collected and transmitted by the BLE-based wireless measurement system (see Supplementary Figs. 5 and 6). The output voltage of the five-stacked I-TENG with a low-pass filter, capacitors (60 μF capacity), and the system load was measured by the oscilloscope (Fig. 4a) and the wireless measurement system (Fig. 4b, c). The stacked I-TENG generated over 4 V under laboratory conditions. However, because of the maximum and minimum voltage limitations of the wireless measurement system, the maximum transmitted voltage signal during the in vivo experiment was 3.75 V, and the transmitted voltage signal in standby mode was 0.5 V. Two packaged devices were inserted by physical surgery subcutaneously in the animal's back and were positioned parallel to the body. Because of differences in the z-axis direction of the devices inserted in the back, the devices showed different energy harvesting properties during the same movement. The normal direction of device 1 is the horizontal direction of the body, and the normal direction of device 2 is the vertical direction of the body. Device 1 is activated by motion more frequently than device 2 due to the distinctive motion behaviors of the mongrel (see Fig. 4b, c). The location of

device 1 was also used for other in vivo experiments. Supplementary Fig. 12 demonstrates real-time energy harvesting performance during daily life when the I-TENG is vertically oriented from the ground and attached to the human chest.

In order to monitor device activity and estimate the daily amount of energy harvested from device 1, we tracked output performance over the course of 24 h (see Fig. 4d). There was more activity during the daytime than at night. Together, the stacked I-TENG generated around 144 mW of power. In addition, we successfully charged low-ESL capacitors, connected in parallel and to various capacitances, that have a capacitance of 60 μF, even based on the small movements that occurred during sleep (see Fig. 4e). When the animal was the most active, the capacitor voltage increased rapidly, and when the animal was calm, the system load consumed energy so that the storage voltage became saturated at a certain level or slowly decreased. Based on the previous preclinical experimental results, we designed the power management system and demonstrated the possibility of charging a battery using biomechanical energy. First, the output power of the I-TENG device charges first energy storage, and the PMIC efficiently transfers the stored energy from the first energy storage to the second energy storage. Figure 4f shows the voltage profile of the second energy storage in the preclinical test. After the second energy storage is charged over 1.8 V, the stored energy is transferred to a Li-ion battery. Therefore, we successfully demonstrate that the stacked I-TENG can recharge a battery for practical purposes inside the body.

The device packaged with the commercial medical purposed biocompatible polymer was implanted subcutaneously in the animal's back, where it remained in constant contact with a muscle layer (see Fig. 5a). In order to verify the biocompatibility of the device, we evaluated the inflammatory response of the muscle layer using Masson's trichrome stain (see Fig. 5b). Figure 5b(i) shows the reference condition of the muscle, in which a few inflammatory cells and a scanty amount of perivascular fibrosis are observed. However, both the encapsulated device and a commercial medical device were associated with a mild inflammatory response and formation of a prominent fibrotic capsule around the device, which does not have any apparent significant differences according to the type of device (see Fig. 5b(ii), 5b(iii), c, Supplementary Fig. 13, and Supplementary Table 1). There was no definite sign of infection during the observation period. The subject exhibited normal daily behavior without severe fever or purulence.

**The self-rechargeable cardiac pacemaker system.** We developed an integrated cardiac pacemaker and TENG power management and battery charging system for a self-rechargeable cardiac pacemaker. Figure 6a presents an optical photo image of a Ti-packaged self-rechargeable cardiac pacemaker. The thickness, from 0.5 to 1.5 mm, of the titanium packaging, ensured biocompatibility and stability, and the lead connector was encapsulated by medical-grade silicon. Supplementary Fig. 14 shows a schematic illustration of the TENG, pacemaker/power management PCB board, and battery structure in the Ti chamber. The as-developed self-rechargeable cardiac pacemaker system received and analyzed atrial and ventricle electrocardiogram (EGM) data, and recharges its battery using the TENG (see Fig. 6b). The block diagram in Fig. 6c represents the overall system operating principle. The atrial and ventricle lead wires sense EGM signals and pace the ventricle when bradycardia occurs, and the TENG harvests energy and charges the battery. In order to demonstrate the pacing performance of the self-rechargeable cardiac pacemaker, we successfully drived asynchronous V pacing mode (VOO) at 150 bpm (see Supplementary Fig. 15), and we also demonstrated VVI mode, which is commonly used for a single-

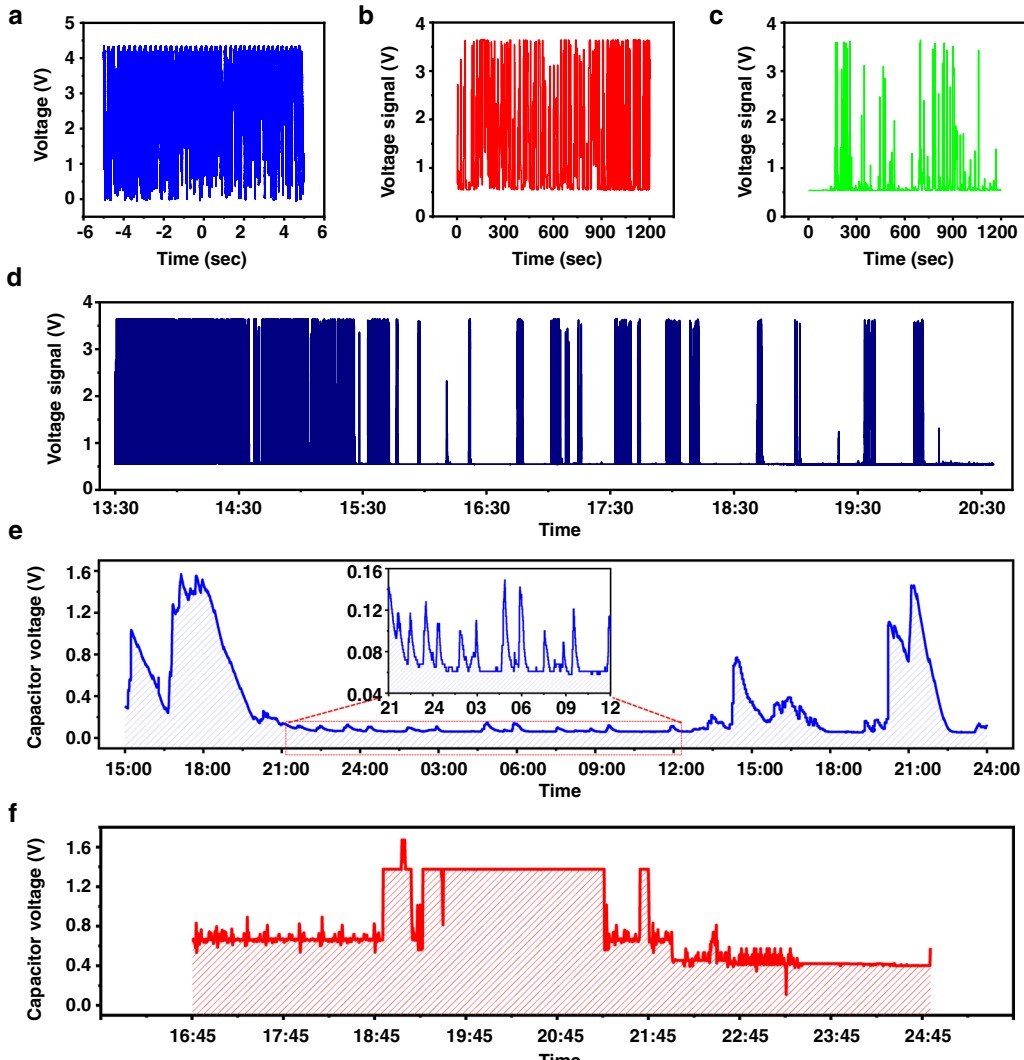

**Fig. 4 In vivo evaluation of the I-TENGs in a large animal preclinical test. a** Output voltage performance of the integrated device measured by an oscilloscope. Transmitted generating voltage information (**b**) from device 1 and (**c**) from device 2. **d** Real-time energy harvesting performance of the I-TENGs during the daytime. **e** Real-time energy charging performance of the first energy storage over the course of one day. **f** Real-time energy charging performance of the second energy storage.

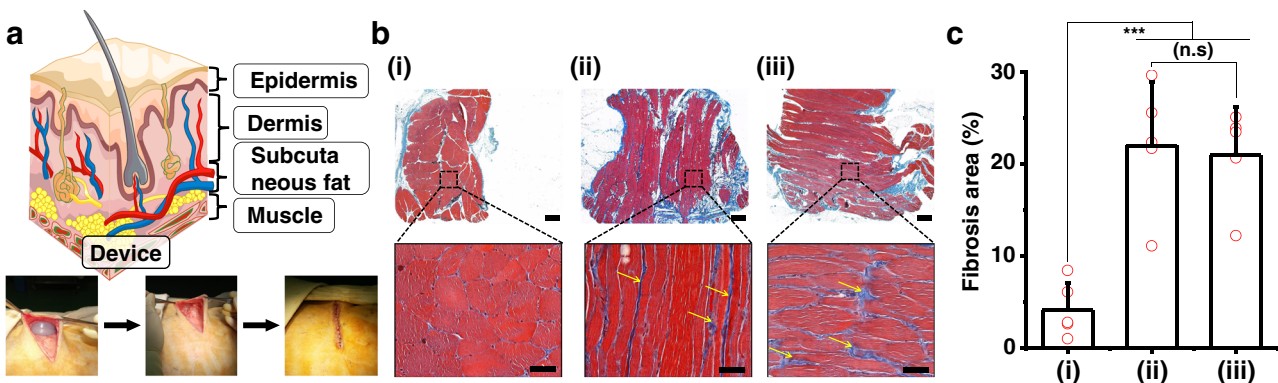

**Fig. 5 Biocompatibility of the encapsulated I-TENG. a** Schematic image of the implanted device under the subcutaneous fat of the mongrel model. **b** Masson's trichrome stain results of (i) no inflammatory cells and scanty amount of perivascular fibrosis observed in normal muscle; (ii) prominent fibrosis and a mild inflammatory response around the encapsulated device; and (iii) similar fibrosis and inflammatory response around a commercial medical device. Red, white, and blue represent normal muscle, fat, and infection, respectively. Yellow arrow was analyzed by randomly selecting each of five groups in sections of tissue implanted. scale bar, 1 mm. Inset figures scale bar, 50 μm. The Masson's trichrome stain results randomly selected five groups of three different conditions. **c** Comparison of fibrosis area (%) of (i) normal muscles, (ii) muscle tissue around the encapsulated device, and (iii) muscle surrounding the commercial medical device, determined using ImageJ software. $n = 5$, $p$ value = 0.0002, one-way ANOVA. Error bars correspond to standard deviations.

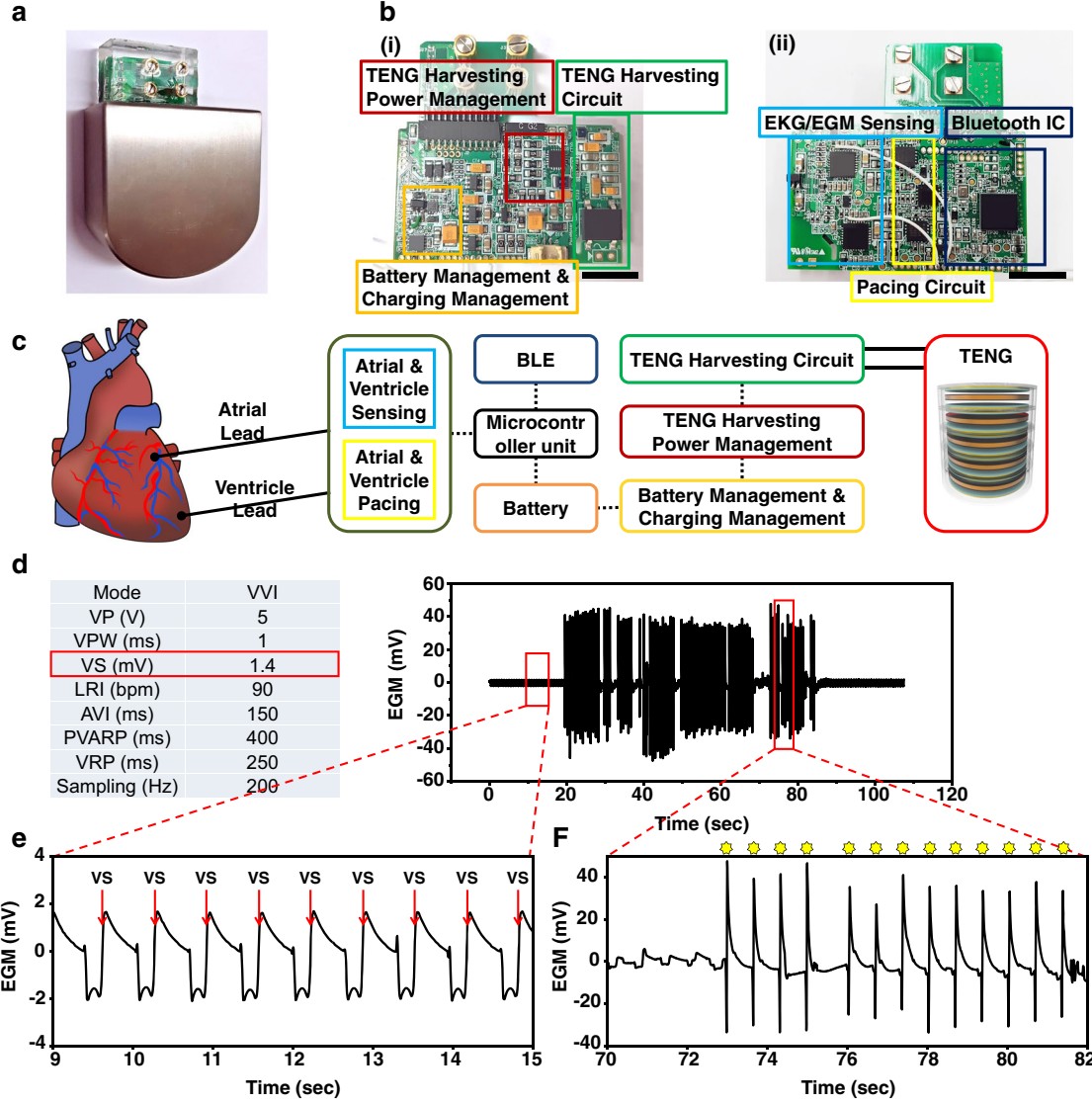

**Fig. 6 Self-rechargeable cardiac pacemaker system. a** Optical image of the self-rechargeable cardiac pacemaker. **b** Photographs of the (i) front and (ii) back of the integrated system. scale bar, 1 cm. **c** Functional block diagram of the self-rechargeable cardiac pacemaker. **d** Ventricular EGM signal from the in vivo animal experiment. **e** Ventricular EGM signals before adenosine injection. **f** Ventricular EGM signals with ventricle pacing after adenosine injection.

chamber pacemaker. Figure 6d–f demonstrates the VVI mode of the self-rechargeable cardiac pacemaker. The ventricle pacing voltage is 5 V with a 1 ms pulse width, the ventricle sensing voltage was 1.4 mV, and the lower rate interval (LRI) of the pacemaker was 90 bpm. The normal heart rate was around 96 bpm; it fell below 90 bpm after adenosine injection. When the heart rate was lower than 90 bpm, the self-rechargeable cardiac pacemaker paced the ventricle and bringed the rate up to 90 bpm. Vagus nerve stimulation is used to more clearly demonstrate bradycardia rather than adenosine injection. We measured electrokardiogram (EKG), atrial EGM, and ventricle EGM data of an adult mongrel in normal conditions (see Supplementary Fig. 16), and caused bradycardia with vagus nerve stimulation (amplitude: 12 mA, pulse width: 1 ms, frequency: 13 Hz). The heart rate decreased from 102 to 80 bpm (see Supplementary Fig. 17), and the self-rechargeable cardiac pacemaker successfully sensed the change and paced the ventricle after the heart rate dropped below 90 bpm (see Supplementary Fig. 18). After vagus nerve stimulation was halted, ventricle pacing was successfully turned off and the normal condition was recovered.

## Discussion

Here, we demonstrated the conversion of mechanical energy into electrical energy inside the body using the I-TENG based on inertia and realized a self-rechargeable cardiac pacemaker system. In contrast to previous TENGs that used direct mechanical deformation to harvest energy, the I-TENG was supplied with sufficient mechanical energy indirectly and could be completely enclosed with body-implantable medical materials. In addition, surface treatments that increase the active area and surface charge density improve current output performance. Under 3 Hz regular vertical motion, the five-stacked I-TENG generated 4.9 $\mu W_{RMS}$/cm$^3$ in the laboratory experiment. Even when multidirectional movements occurred concurrently, satisfying a minimum normal-direction displacement of the freestanding unit secured the device performance. Preclinical testing in a large animal experiment demonstrated the significant power performance of the I-TENG driven by biomechanical energy and inertia. The I-TENG inserted in the body was also analyzed and monitored in real-time. Finally, the self-rechargeable cardiac pacemaker

successfully demonstrated VOO and VVI mode using brady-cardia in a mongrel. The power performance differed between the laboratory and preclinical experiments because of differences in input mechanical energy or animal motion. Future fabrication efforts might incorporate the better power performance of TENGs into self-powering body-implantable medical devices and health-monitoring systems to advance the welfare of patients.

## Methods

**I-TENG fabrication**. A 50 μm PFA film (10 cm × 10 cm) was physically and electrically treated to modify its surface. A reactive ion etching (RIE) system (SNTEK RIE5000) was used for the physical treatment: reactive plasma was irradiated with 50 standard cubic centimeters per minute (SCCM) of oxygen ($O_2$) gas and 50 SCCM of argon (Ar) gas at a radio frequency power of 100 W under 0.2 Torr pressure for 5 min. After the RIE process, the treated PFA film was placed on a Cu plate, and needles were installed ~1 cm above the PFA film. We connected needles to the cathode and connected the Cu plate to the ground. A polarization voltage of 15 kV was applied for 15 min. Next, we inverted the PFA film and reverse-connected the cathode and ground. To the reversed PFA film, a polarization voltage of 15 kV was applied for 15 min again. The prepared PFA film was attached to the Cu mass (thickness, 0.8 mm; radius, 1.25 cm) using a carbon double side tape as an adhesive layer, and prepared PVA-NH₂ solution[36] was spin-coated on the Au-deposited substrate (radius, 1.5 cm). A 2 mm thick acrylic layer was used as a gap between the top and bottom substrates. In order to encapsulate the I-TENG, it was coated with a commercial biocompatible polymer (C6-540 Liquid Silicone Rubber, Dow Corning) with a thickness of 5 mm. The encapsulated device was chemically sterilized and rinsed for animal experiments.

**TENG characterization**. The preparation of the I-TENGs has described in the subsection "Methods: I-TENG fabrication." A shaker (Model No. ET-126B-4, Labworks Inc.) was used to apply regular vertical displacement to the I-TENGs. A Tektronix DPO 3052 digital phosphor oscilloscope and low-noise current pre-amplifier (Model No. SR570, Stanford Research Systems, Inc.) were used for electrical measurements. Origin 2018 used for TENG electrical output data analysis. The study was approved by the Institutional Review Board of the Seoul National University Hospital, Seoul, Korea (#2104-230-1217). The authors affirm that human research participants provided informed consent for publication of the images in Fig. 1a.

**Animal preparation and characterization**. Three adult mongrels (25–30 kg) were purchased from the International Center Laboratory Animal and were housed in the Seoul National University Hospital Biomedical Research Institute. General anesthesia was induced with Zoletil® (intravenously injected, 5 mg/kg; a combination of tiletamine/zolazepam, Virbac S/A, Carros, France) and maintained with isoflurane gas (1–2% in $O_2$). After induction, the animals were intubated and mechanically ventilated. Core body temperature was maintained at 36.5–37 °C, and a limb lead electrocardiogram was continuously monitored during the procedure. Under sterile surgical conditions, the packaged device was implanted subcutaneously in the back area. After the surgical procedure, meloxicam was intravenously injected (0.2 mg/kg) for one day to relieve pain and discomfort. After 1 week of recovery, recordings were made while the mongrel was ambulatory. The protocol for this study was approved by the Institutional Animal Care and Use Committee of the Seoul National University Hospital, and the animals were maintained in the facility accredited by AALAC International (#001169) in accordance with the Guide for the Care and Use of Laboratory Animals, eighth edition, NRC (2010). EKG, EGM data were analyzed using LabChart 8. ImageJ and Graphpad ver 6.0 were used to analyze Masson's trichrome stain results.

**The Masson's trichrome staining procedure**. Masson's trichrome staining assay was conducted to evaluate the inflammatory response of the muscle layer using a trichrome stain kit (ab150686, Abcam, US). Sections are deparaffinized and hydrated in distilled water. The slide is placed in 56–64 °C preheated Bouin's fluid (60 min) followed by a cooling period (10 min). The slide is rinsed until the section is clear by water. The sections are stained with working Weigert's iron hematoxylin for 5 min. After rinsing the slide, Biebrich scarlet/acid fuchsin solution is applied to the slide for 15 min. After rinsing the slide, the section is differentiated in phosphomolybdic/phosphotungstic acid solution until the collagen is not red. Aniline blue solution is applied to the slide for 5–10 min. After rinsing the slide, acetic acid solution (1%) is applied to the slide for 3–5 min. Samples are dehydrated in two changes of 95% alcohol.

**Reporting summary**. Further information on research design is available in the Nature Research Reporting Summary linked to this article.

## Data availability

All relevant data supporting the key findings of this study are available within the article and its Supplementary Information files or from the corresponding author upon reasonable request. Source data are provided with this paper. A reporting summary for this article is available as a Supplementary Information file. Source data are provided with this paper.

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

## Acknowledgements
This work was supported by the Nano Material Technology Development Program (2020M3H4A1A03084600) and the Basic Science Research Program (2021R1A2C2010990) through the National Research Foundation of Korea (NRF) funded by the Ministry of Science and ICT.

## Author contributions
S.-W.K., H.R., and H.-M.P. conceived the idea. H.R., H.-M.P., H.S.M., T.Y.K., H.-J.Y., S.S.K., J.K., and B.K. fabricated, measured, and simulated the devices. M.-K.K. and E.-K.C. performed the in vivo experiments. S.-W.K., T.H.H., and E.-K.C. commented on the research outcomes. H.R., H.-M.P., E.-K.C., and S.-W.K. analyzed the data and wrote the manuscript. S.-W.K. supervised the overall conception and design of this project. All authors contributed to the discussion on the results and improved the manuscript.

## Competing interests
H.R., T.Y.K., H.-J.Y., S.S.K., J.K., and S.-W.K. are co-inventors of the patent entitled "ENERGY GENERATOR USING INERTIA" patent number: 10-1940911, Republic of Korea. The other authors declare no competing interests.
