## [Peer Review File · Nature Communications]

REVIEWER COMMENTS

Reviewer #1 (Remarks to the Author):

This manuscript reports a inertial-driven triboelectric nanogenerator (I-TENG) and a self-chargeable cardiac pacemaker system. A five-stacked I-TENG is fabricated to boost up the output current, which produces an output of $4.9 \mu\text{W}/\text{cm}^3$ (root mean-square output) under a 3Hz shaking condition. The I-TENG is encapsulated for in-vivo evaluation, which was implanted subcutaneously in an adult mongrel's back. Lastly, the I-TENG was integrated with a cardiac pacemaker, demonstrating the self-rechargeable cardiac pacemaker system. The authors should address the following comments before this work is considered to be accepted.

1. Why did the authors choose the PFA and PVA-NH₂ layers as the negative and positive triboelectric materials? Any specific property of them and how is the performance compared with common triboelectric materials such as PTFE?
2. What is the minimum force or acceleration in order to make the PFA/Cu/PFA freestanding unit fully contact with the upper triboelectric layer? In addition, how is the performance of the device varies with the applied force/acceleration or frequency?
3. This implanted device seems to be non-flexible, and the thickness of the five-stack layers would be as large as 12 mm. Would that cause a non-negligible discomfort to the subject considering the large thickness and the un-matched mechanical properties?
4. How is the operating efficiency calculated in Figure 2h?
5. Figure 4d presents the output voltage from device 1 or device 2? Or both? It should be explained in the main text.
6. Device 1, which is vertical to the body, can harvest more energy than device 2 as appeared in the testing data from Figure 4b-c. Does this mean that the I-TENG has an optimum implanted position regarding the maximum energy collected from body motions? And apparently, it is better to be placed vertically in the mongrel model, perhaps due to the distinctive motion behaviors of the mongrel compared to the human. The authors should discuss more on the placement of the I-TENG and add more explanation to the experimental results in Figure 4 b-c.
7. Which device is used to charge up the capacitors?
8. When calculating the input energy (equation 2), the authors used the weight of the freestanding unit. Wouldn't it be more accurate to adopt the whole mass of the device?
9. The photo of the five-stacked I-TENG should be provided for a better illustration of the device.
10. How much energy can be stored in the li-ion battery each time?
11. How long does it take for the I-TENG to charge up the battery for the effective operation of the pacemaker? For example, 1-hour charging time can sustain the pacemaker work for how long?
12. To provide good background about the TENG technology for implantable applications, authors may consider to include the next few works in the introduction as the references. Nano Energy, vol. 65, 104039, 2019; Adv. Sci., vol. 6, no. 14, 1900149, 2019; ACS Nano, vol. 13, no. 3, pp. 3589-3599, 2019.

Reviewer #2 (Remarks to the Author):

In this manuscript, the authors reported a self-rechargeable cardiac pacemaker system with the commercial coin battery-sized high-performance inertia-driven in vivo triboelectric nanogenerator (i-TENG) based on body motion and gravity. The maximum volume power density of the I-TENG was $4.9 \mu\text{WRMS}/\text{cm}^3$ at a load resistance of $\sim 10 \text{ M}\Omega$ in a laboratory experiment. The authors demonstrated that the device harvested around 144 mW during the daytime in the preclinical large animal experiment. Furthermore, the authors successfully integrated a cardiac pacemaker with the i-TENG

and demonstrated VVI mode of the self-rechargeable cardiac pacemaker. This work represent a quite interesting and promising in vivo energy harvester for low-power implantable electronic devices. Before accepting the manuscript for publication, some explains are needed to strengthen authors' statements.

1. The i-TENG harvest biomechanical energy based on body motion (inertia) and gravity. The movement mode of the device shown by the authors in the previous part of the article is vertical movement (Fig. 1b and 1c, Fig. 2a, and Fig. 6c). However, When the i-TENG is integrated with the pacemaker, its movement mode is horizontal (Fig. S10). Compared with animals, humans walk upright. After the device is implanted in the body, the movement mode changes and the performance will change greatly. So how is it placed during the actual implantation process? Please give more explanation.
2. What are the data on the current and charge of the device in vivo? These are very important for evaluating the performance of i-TENG. In addition, how does the device realize energy storage and release?
3. The authors need to evaluate how long the device collects biomechanical energy to satisfy an effective pacing in vivo.
4. Line 96-97-"...Fig. 1b shows a schematic image of a cylindrical I-TENG with a radius of 1.5 cm and a height of 2.4 mm that is used to harvest vertical body motion....". Is the device height description right?
5. What is the relationship between the "144 mW" and the "20 % of the time" (Line 78-80)? As we know, power is not related to time. The reviewer thinks that the values of energy (Joule) or charges (Coulomb) should be given here.
6. The frequency of 3 Hz (Line 141-143) seems too high for normal human moving (e.g., walking) or organ movements (e.g., heartbeat). Therefore, the frequency characteristic of the I-TENG might restrict the practical application of itself directly. The authors should consider the frequency characteristic of the I-TENG more deeply. By the way, the 3 Hz was the best frequency for the device?
7. The size of the I-TENG was similar to a commercial coin battery; meanwhile, the weight of the device was also important to a cardiac pacemaker. Furthermore, the authors should discuss the influence of the weight of the I-TENG on cardiac pacemakers.
8. The manuscript states "that the maximum volume power density of the I-TENG was 4.9 $\mu\text{WRMS}/\text{cm}^3$ at a load resistance of $\sim 10 \text{ M}\Omega$ in a laboratory experiment, which is a significantly higher output power than that of fully encapsulated in vivo TENGs" is misleading. The author should read and compare the references and this work carefully.
9. The in vivo output performance of I-TENG requires more details, such as the provenance details within a biomotion cycle. In addition, the charge-discharge curves of I-TENG to energy storage devices is required.
10. The I-TENG or in vivo TENG can easily be confused with iTENG, and it is recommended that thses type of implantable TENG use iTENG uniformly.

Reviewer #3 (Remarks to the Author):

The manuscript entitled "Self-rechargeable cardiac pacemaker system with triboelectric nanogenerators" reported an implantable TENG generating sufficient power for implantable medical devices (IMDs) by harvesting inertia energy. In most IMDs, battery with limited lifetime demands retrieval procedures that are usually invasive and cause infections/inflammations. Self-sufficient devices driven by biomechanical energy represent a promising alternative to battery to avoid secondary operations. The significance of this work is high, and it also has well-organized figures with a clear logic. I would recommend its publication to the journal of Nature Communications if the authors could clarify the following issues that appeared in the current manuscript:

1. Even through a parallel-connection, the impedance of the I-TENG around 10 M Ω is still large, significantly higher than the impedance of batteries and IMDs. Although not calculated here, the charging efficiency might still be a problem. A lot of energy might be lost during the process of charging up different energy storage units. Please elaborate more about the charging efficiency.
2. The authors mentioned the energy harvested from animal could eventually charge up a Li battery. There is no data related to Li battery charging neither in the manuscript nor in the supporting information. Only 1st and 2nd energy storage modules charging data was found. The authors should add relevant data and explain.
3. Regarding the pacemaker application, this I-TENG could generate ~ 100 V in vitro output whereas the in vivo output is only a few volts. Given the in vitro power density is 4.9 $\mu\text{W}/\text{cm}^3$, the in vivo value should be significantly lower. Nevertheless, a commercial pacemaker usually requires 10-20 μW , which is hardly met by the i-TENG harvesting in vivo energy, raising a concern for its practical implantation. This discrepancy needs to be discussed. Meanwhile, the authors should specify how the device charged up the pacemaker. I assume it is done by the electrodynamic transducer instead of in vivo biomechanical energy.
4. The authors noticed the synchronization of different units' movement is of great importance for the output, considering the phases of electricity. They claimed that the design enables mechanical energy applied simultaneously. How did the design affect the mechanical energy input? It is not clear and should be elaborated more.
5. The authors used amine-functionalized poly(vinyl alcohol) (PVA-NH₂) and perfluoroalkoxy (PFA) as triboelectric materials. Why the authors use these two materials and what are the advantages of these two materials compared to other commonly used ones?

Reviewer #1 (Remarks to the Author):

This manuscript reports a inertial-driven triboelectric nanogenerator (I-TENG) and a self-chargeable cardiac pacemaker system. A five-stacked I-TENG is fabricated to boost up the output current, which produces an output of $4.9 \mu\text{W}/\text{cm}^3$ (root mean-square output) under a 3Hz shaking condition. The I-TENG is encapsulated for in-vivo evaluation, which was implanted subcutaneously in an adult mongrel's back. Lastly, the I-TENG was integrated with a cardiac pacemaker, demonstrating the self-rechargeable cardiac pacemaker system. The authors should address the following comments before this work is considered to be accepted.

Our response: Authors thank the reviewer for these positive comments and for the reviewer's thorough examination of the text. We provide responses below, accompanied by corresponding changes to the manuscript. Changes to the manuscript appear in blue, highlighted text.

Comment 1: Why did the authors choose the PFA and PVA-NH₂ layers as the negative and positive triboelectric materials? Any specific property of them and how is the performance compared with common triboelectric materials such as PTFE?

Response 1: Thank you for the reviewer's comment. In order to increase output power performance of the I-TENG, we utilized PFA instead of PTFE as a negative triboelectric material because of the higher negative triboelectric properties of PFA than PTFE (*ACS Appl. Mater. Interfaces* **2016**, 8, 18519–18525). We used PVA-NH₂, which has a large number of H atoms, as a positive triboelectric material to facilitate higher positive triboelectric property than PVA (*Energy Environ. Sci.*, **2019**, 12, 3156). Although metals are well-known positive triboelectric materials, a metal electrode can increase the potential electrical short-circuit problem between top/bottom metal electrodes and the freestanding Cu electrode. The spin coating process facilitates a micron-thick coating of PVA-NH₂ on the substrate and strong mechanical stability between the substrate and PVA-NH₂. Therefore, we utilized PFA and PVA-NH₂ as triboelectric materials instead of PTFE and metal.

Comment 2: What is the minimum force or acceleration in order to make the PFA/Cu/PFA freestanding unit fully contact with the upper triboelectric layer? In addition, how is the performance of the device varies with the applied force/acceleration or frequency?

Response 2: We appreciate the reviewer's valuable comment. We applied 0.4 N to the I-TENG to achieve full contact between the PFA/Cu/PFA freestanding unit and the upper triboelectric layer. The applied force is a critical factor to decide input energy per each cycle, and input frequency decides total input energy. If we apply larger force, the I-TENG will generate larger power due to increasing the input mechanical energy. When we change the frequency with a similar input force, maximum output voltages are similar with different output frequencies (see Figure R1). We added output voltage performance of the I-TENG depending on various frequencies in the manuscript.

Figure R1. Output voltage performance of the I-TENG as a function of a frequency range from 0.5 to 3 Hz.

Modifications to manuscript: On page 6, line 145 in the manuscript, we added “ The five stacked I-TENGs generated similar voltage under different frequencies (see Supplementary Fig. S8).” We also added Supplementary Fig. S8 in the Supplementary Materials.

Comment 3: This implanted device seems to be non-flexible, and the thickness of the five-stack layers would be as large as 12 mm. Would that cause a non-negligible discomfort to the subject considering the large thickness and the un-matched mechanical properties?

Response 3: We agree with the referee’s comment. We have tried to prevent any discomfort to the subject during the experiments. Although the implanted device is thicker than commercial pacemakers (e.g. 7.5 mm in thick; Ingenio K172,173,714, Boston Scientific, 6 mm in thick; ASSURITY MRI, Abbott), no uncomfortable behavior was found in the subject during the experiments.

Comment 4: How is the operating efficiency calculated in Figure 2h?

Response 4: Thank you for the referee’s comment. If you are asking about the energy conversion efficiency of the I-TENG, we provided a detailed description in Supplementary Note S4. Based on the calculation, the I-TENG has a 0.235% energy conversion efficiency. If you are asking about energy conversion efficiency in Figure 3h, we calculate the energy conversion efficiency of the stacked I-TENGs depending on the working angles and displacements (see Figure R2). Most conditions have a low energy conversion efficiency of less than 0.1% due to insufficient input force or multi-directional movements that can impact the acrylic gap spacer instead of the top triboelectric layer. On the other hand, when a working angle is 90°, every condition shows over 0.235% efficiency due to only a z-axis directional movement. We added additional energy conversion efficiency data in the manuscript.

Figure R2. The energy conversion efficiency of the I-TENG depending on different displacement and working angle conditions.

Modifications to manuscript: On page 8, line 194 in the manuscript, we added “Supplementary Fig. S11 shows the energy conversion efficiency of each experimental condition.”. We also added Supplementary Fig. S11 in the Supplementary Materials.

Comment 5: Figure 4d presents the output voltage from device 1 or device 2? Or both? It should be explained in the main text.

Response 5: We appreciate the reviewer’s comment. Figure 4d presents the output voltage from device 1, and the same place used for other *in vivo* experiments. We modify the manuscript based on the reviewer’s comment.

Modifications to manuscript: On page 9, line 221 in the manuscript, we modified “~ the daily amount of energy harvested from the device 1, ~”.

Comment 6: Device 1, which is vertical to the body, can harvest more energy than device 2 as appeared in the testing data from Figure 4b-c. Does this mean that the I-TENG has an optimum implanted position regarding the maximum energy collected from body motions? And apparently, it is better to be placed vertically in the mongrel model, perhaps due to the distinctive motion behaviors of the mongrel compared to the human. The authors should discuss more on the placement of the I-TENG and add more explanation to the experimental results in Figure 4 b-c.

Response 6: We agree with the referee’s comment. Output performance of the I-TENG is related to the implanted position, and it had an optimum position regarding subject behavior. As a reviewer mentioned, we added additional descriptions in the manuscript.

Modifications to manuscript: Page 9, line 217 in the manuscript, we added “~ due to the distinctive motion behaviors of the mongrel”.

Comment 7: Which device is used to charge up the capacitors?

Response 7: Thank you for your comment. Since we found the optimum position (position of the device 1), we inserted a device into a similar position to achieve better performance. Every I-TENGs show similar performance under laboratory conditions. We add an additional description in the manuscript.

Modifications to manuscript: On page 9, line 217 in the manuscript, we added “The location of device 1 was also used for other *in vivo* experiments.”.

Comment 8: When calculating the input energy (equation 2), the authors used the weight of the freestanding unit. Wouldn't it be more accurate to adopt the whole mass of the device?

Response 8: We appreciate the reviewer’s comment. To define energy conversion efficiency in the manuscript, we considered active mass and applied force to calculate input energy. We modified the mass condition from the freestanding units to the whole mass of the device. The total mass of the device is ≈ 28.5 g, and the calculated energy conversion efficiency is 0.235%. We modified that manuscript accordingly.

Modifications to manuscript: On Page 7, line 157-159 in the manuscript, we modified the values of input kinetic energy and internal energy conversion efficiency. We also modified the values of Supplementary Note S4 in the Supplementary Materials.

Comment 9: The photo of the five-stacked I-TENG should be provided for a better illustration of the device.

Response 9: We added additional photo images of the five-stacked I-TENG with different angles according to the reviewer’s recommendation.

Figure R3. Photo images of the five-stacked I-TENGs from (a) top, (b) tilted, and (c) side views.

Modifications to manuscript: On page 5, line 106 in the manuscript, we added “~ and Supplementary Fig. S3 show ~”, and added Fig. S3 in the Supplementary Materials.

Comment 10: How much energy can be stored in the li-ion battery each time?

Response 10: We appreciate the reviewer’s comment. $662 \mu\text{W}$ can be stored in the Li-ion battery.

Comment 11: How long does it take for the I-TENG to charge up the battery for the effective operation of the pacemaker? For example, 1-hour charging time can sustain the pacemaker work for how long?

Response 11: Thank you for the valuable comment. For example, if the I-TENG operates 3 Hz for 1-hour, the total output power is 411.48 mW. If every energy storage without any loss, a commercial pacemaker, which requires 10-20 μ W, can operate 5 to 11 hours.

Comment 12: To provide good background about the TENG technology for implantable applications, authors may consider to include the next few works in the introduction as the references. Nano Energy, vol. 65, 104039, 2019; Adv. Sci., vol. 6, no. 14, 1900149, 2019; ACS Nano, vol. 13, no. 3, pp. 3589-3599, 2019.

Response 12: Thank you for suggesting recently published papers. We added recommended references in the revised manuscript.

Reviewer #2 (Remarks to the Author):

In this manuscript, the authors reported a self-rechargeable cardiac pacemaker system with the commercial coin battery-sized high-performance inertia-driven in vivo triboelectric nanogenerator (i-TENG) based on body motion and gravity. The maximum volume power density of the I-TENG was $4.9 \mu\text{WRMS}/\text{cm}^3$ at a load resistance of $\sim 10 \text{ M}\Omega$ in a laboratory experiment. The authors demonstrated that the device harvested around 144 mW during the daytime in the preclinical large animal experiment. Furthermore, the authors successfully integrated a cardiac pacemaker with the i-TENG and demonstrated VVI mode of the self-rechargeable cardiac pacemaker. This work represent a quite interesting and promising in vivo energy harvester for low-power implantable electronic devices. Before accepting the manuscript for publication, some explains are needed to strengthen authors' statements.

Our response: We thank the reviewer for these positive comments and for the reviewer's thorough examination of the text. We provide responses below, accompanied by corresponding changes to the manuscript. Changes to the manuscript appear in blue, highlighted text.

Comment 1: The i-TENG harvest biomechanical energy based on body motion (inertia) and gravity. The movement mode of the device shown by the authors in the previous part of the article is vertical movement (Fig. 1b and 1c, Fig. 2a, and Fig. 6c). However, When the i-TENG is integrated with the pacemaker, its movement mode is horizontal (Fig. S10). Compared with animals, humans walk upright. After the device is implanted in the body, the movement mode changes and the performance will change greatly. So how is it placed during the actual implantation process? Please give more explanation.

Response 1: We appreciate the reviewer's valuable comment. Fig. S14 demonstrated a schematic image of a self-rechargeable cardiac pacemaker for animal experiments. As you already pointed out, the orientation of the I-TENG should be modified for a human clinical test. Because cardiac pacemakers are usually placed in the chest, the orientation of the I-TENG should rotate 90 degrees for vertical movements. We tested the possibility of the I-TENG for human movement conditions. The I-TENG is vertically oriented from the ground, and attached to the chest. While a subject walking, running, climbing stairs, and resting, the I-TENGs successfully charged $60 \mu\text{F}$ capacitors (see Figure R4). This result indicates that simple orientation modification of the I-TENG can store the biomechanical energy of the human. We modify the manuscript based on your comments.

Figure R4. Real-time energy harvesting performance of the I-TENGs while walking, running, climbing stairs, and resting.

Modifications to manuscript: On page 9, line 218-220 in the manuscript, we added “Supplementary Fig. S12 demonstrates real-time energy harvesting performance during daily life when the I-TENG is vertically oriented from the ground and attached to the human chest.”.

Comment 2: What are the data on the current and charge of the device in vivo? These are very important for evaluating the performance of i-TENG. In addition, how does the device realize energy storage and release?

Response 2: Thank you for the reviewer’s comment. We believe that output performance of the I-TENG is the same as in both *in vivo* and laboratory conditions. Unfortunately, the wireless measurement system only can measure voltage, not current. Figure S5 shows the system-level block diagram of the power management system. Since energy stores in 1st energy storage (60 μ F capacitance) over 1.5 V, PMIC automatically transfers stored energy to the 2nd energy storage (150 μ F capacitance). After the 2nd energy storage is charged over 1.8 V, the stored energy was transferred to a Li-ion battery.

Comment 3: The authors need to evaluate how long the device collects biomechanical energy to satisfy an effective pacing in vivo.

Response 3: We appreciate the reviewer’s comment. The I-TENG only provides energy for charging Li-ion battery, and the energy source for the pacing function is Li-ion battery. Thus, we do not evaluate the charging time for pacing.

Comment 4: Line 96-97-“...Fig. 1b shows a schematic image of a cylindrical I-TENG with a radius of 1.5 cm and a height of 2.4 mm that is used to harvest vertical body motion....”. Is the device height description right?

Response 4: Thank you for the reviewer’s comment. Yes, the dimension of a cylindrical I-TENG is correctly described in the manuscript.

Comment 5: What is the relationship between the “144 mW” and the “20% of the time” (Line 78-80)? As we know, power is not related to time. The reviewer thinks that the values of energy (Joule) or charges (Coulomb) should be given here.

Response 5: We appreciate the reviewer’s comment, and apologize for making misunderstanding in our explanation. Compared to the total experiment time, I-TENG only activates less than 20% of the experiment time. During the activation time, the I-TENG generated 144 mW. We modify the main text accordingly.

Modifications to manuscript: On page 4, line 76-78 in the manuscript, we modified “The device harvested around 144 mW in the preclinical large animal experiment, and the activity time of the stacked I-TENG is less than 20% of the total experiment time due to the behavior of the animal in its cage.”.

Comment 6: The frequency of 3 Hz (Line 141-143) seems too high for normal human moving (e.g., walking) or organ movements (e.g., heartbeat). Therefore, the frequency characteristic of the I-TENG might restrict the practical application of itself directly. The authors should consider the frequency characteristic of the I-TENG more deeply. By the way, the 3 Hz was the best frequency for the device?

Response 6: The authors appreciate the reviewer’s valuable comment. Because the I-TENGs do not have a resonance frequency, the applied force is more critical than applied frequencies to operate the I-TENGs. When we change the frequency with a similar and sufficient applied force, maximum output voltages are similar with different output frequencies (see Figure R5). Thus, any frequency conditions are suitable for the I-TENG. We added output voltage performance of the I-TENG depending on various frequencies.

Figure R5. Output voltage performance of the I-TENG as a function of a frequency range from 0.5 to 3 Hz.

Modifications to manuscript: On page 6, line 145 in the manuscript, we added “The five stacked I-TENGs generated similar voltage under different frequencies (see Supplementary Fig. S8).” We also added Supplementary Fig. S8 in the Supplementary Materials.

Comment 7: The size of the I-TENG was similar to a commercial coin battery; meanwhile, the weight of the device was also important to a cardiac pacemaker. Furthermore, the authors should discuss the influence of the weight of the I-TENG on cardiac pacemakers.

Response 7: Thank you for the reviewer’s comment. The weight of the single I-TENG (thickness in 2.4 mm) is 5.7 g, which is a similar weight to a commercial non-rechargeable CR3032 coin battery (thickness in 3.2 mm; 6.8 g). We do not recognize any uncomfortable behaviors from the subject during the experiment, and the I-TENG can adapt to the cardiac pacemaker without a huge mass increase.

Comment 8: The manuscript states “that the maximum volume power density of the I-TENG was 4.9 μ WRMS/cm³ at a load resistance of \sim 10 M Ω in a laboratory experiment, which is a significantly higher output power than that of fully encapsulated in vivo TENGs” is misleading. The author should read and compare the references and this work carefully.

Response 8: We appreciate the reviewer’s comment. We carefully compared the references and modified the manuscript.

Modifications to manuscript: On page 4, line 71-73, we modify “ The maximum volume power density of the I-TENG was $4.9 \mu\text{W}_{\text{RMS}}/\text{cm}^3$ at a load resistance of $\sim 10 \text{ M}\Omega$ in a laboratory experiment, which is a competitive high output performance¹¹⁻¹⁷. ”.

Comment 9: The *in vivo* output performance of I-TENG requires more details, such as the provenance details within a biomotion cycle. In addition, the charge-discharge curves of I-TENG to energy storage devices is required.

Response 9: The authors appreciate the reviewer’s comment. Figure S2 provides a detailed working mechanism of the I-TENG. When a subject moves upward and all parts move upward together, the freestanding unit does not detach from the bottom surface (Figure S2a). After a subject movement stops/changes, the freestanding unit moves upward by inertia and contacts with the top triboelectric layer (Figure S2c). By gravity, the freestanding unit moves downward and contacts the bottom triboelectric layer (Figure S2d).

Figure R6 demonstrates capacitors charging and battery charging behavior using the I-TENG and PMIC under the laboratory experimental conditions because *in vivo* environment is difficult to precisely analyze the charge-discharge properties of energy storage devices. The voltage set-up of PMIC adjusts for laboratory experimental conditions. The 1st energy storage (60 μF) charges to 1.4 V and discharges until 1.25 V to transfer energy into 2nd energy storage (150 μF , Figure R6a). The 2nd energy storage charges to 3.4 V and discharges until 2.15 V to charge a small battery (1 mAh capacity) for demonstration (Figure R6b). The battery charges from 2 V to 2.5 V after 1.5 hours (Figure R6c). We revised that manuscript according to the reviewer’s recommendation.

Figure R6. Charging behaviors of (a) 1st and (b) 2nd energy storages in PMIC and (c) battery by the stacked I-TENG.

Modifications to manuscript: On page 7, line 161-170, we added “Based on the laboratory experimental results, we evaluated the power management system and demonstrated a small battery (ML414H, Seiko Instruments Inc., 1 mAh capacity) charging using by the five-stacked I-TENGs. First, the output power of the I-TENG device charges parallel-connected low-equivalent series inductance (ESL) capacitors with a capacitance of 60 μF (*i.e.*, 1st energy storage) after filtering through a low-pass filter (see Fig. 2g). Then, in order to manage the high voltage of the 1st energy storage, the 1st energy storage discharges to larger energy storage with the help of a PMIC. The larger energy storage that consists of parallel-connected low-equivalent series resistance capacitors with a capacitance of 150 μF (*i.e.*, 2nd energy storage) could store

the same amount of energy at most 0.4 times lower voltage (see Fig. 2h). After the 2nd energy storage is charged up to 3.4 V, the stored energy transfers to a Li-ion battery (see Fig. 2i).” We also revised main Figure 2 on page 19.

Comment 10: The I-TENG or in vivo TENG can easily be confused with iTENG, and it is recommended that thses type of implantable TENG use iTENG uniformly.

Response 10: We appreciate the reviewer’s comment. We carefully revised the manuscript based on the reviewer’s comment.

Reviewer #3 (Remarks to the Author):

The manuscript entitled “Self-rechargeable cardiac pacemaker system with triboelectric nanogenerators” reported an implantable TENG generating sufficient power for implantable medical devices (IMDs) by harvesting inertia energy. In most IMDs, battery with limited lifetime demands retrieval procedures that are usually invasive and cause infections/inflammations. Self-sufficient devices driven by biomechanical energy represent a promising alternative to battery to avoid secondary operations. The significance of this work is high, and it also has well-organized figures with a clear logic. I would recommend its publication to the journal of Nature Communications if the authors could clarify the following issues that appeared in the current manuscript:

Our response: We thank the reviewer for these positive comments and for the reviewer’s thorough examination of the text. We provide responses below, accompanied by corresponding changes to the manuscript. Changes to the manuscript appear in blue, highlighted text.

Comment 1: Even through a parallel-connection, the impedance of the I-TENG around 10 M Ω is still large, significantly higher than the impedance of batteries and IMDs. Although not calculated here, the charging efficiency might still be a problem. A lot of energy might be lost during the process of charging up different energy storage units. Please elaborate more about the charging efficiency.

Response 1: Thank you for the referee’s comment. If you are asking about the energy conversion efficiency of the I-TENG, we provided a detailed description in Supplementary Note S4. Based on the calculation, the I-TENG has a 0.235% energy conversion efficiency. If you are asking about energy conversion efficiency in Figure 3h, we calculate the energy conversion efficiency of the stacked I-TENGs depending on the working angles and displacements (see Figure R7). Most conditions have a low energy conversion efficiency of less than 0.1% due to insufficient input force or multi-directional movements that can impact the acrylic gap spacer instead of the top triboelectric layer. On the other hand, when a working angle is 90°, every condition shows over 0.235% efficiency due to only a z-axis directional movement. We added additional energy conversion efficiency data in the manuscript.

Figure R7. The energy conversion efficiency of the I-TENG depending on different displacement and working angle conditions.

Modifications to manuscript: On page 8, line 194 in the manuscript, we added “Supplementary Fig. S11 shows the energy conversion efficiency of each experimental condition.”. We also added Supplementary Fig. S11 in the Supplementary Materials.

Comment 2: The authors mentioned the energy harvested from animal could eventually charge up a Li battery. There is no data related to Li battery charging neither in the manuscript nor in the supporting information. Only 1st and 2nd energy storage modules charging data was found. The authors should add relevant data and explain.

Response 2: We appreciate the reviewer’s valuable comment. It is very difficult to monitor the voltage level of the battery during the *in vivo* experiment, and it is also difficult to demonstrate a large battery (270 mAh capacity) charging behavior using the I-TENG in a short period. Instead, we demonstrate the charging properties of the 1st and 2nd energy storage and battery under laboratory experimental conditions. The voltage set-up of PMIC adjusts for laboratory experimental conditions. The 1st energy storage (60 uF) is charged to 1.4 V and discharge until 1.25 V to transfer energy into 2nd energy storage (150 uF, Figure R8a). The 2nd energy storage is charged to 3.4 V and discharges until 2.15 V to charge a small battery (1 mAh capacity) for demonstration (Figure R8b). The battery charges from 2 V to 2.5 V during 1.5 hours (Figure R8c). We modified the description in the manuscript.

Figure R8. Charging behaviors of (a) 1st and (b) 2nd energy storages in PMIC and (c) battery by I-TENG.

Modifications to manuscript: On page 19, we revised main Figure 2. On page 9, line 228-235, we modified “Based on the previous preclinical experimental results, we designed the power management system and demonstrated the possibility of charging a battery using biomechanical energy. First, output power of the I-TENG device charges 1st energy storage, and the PMIC efficiently transfers the stored energy from the 1st energy storage to the 2nd energy storage. Fig. 4f shows the voltage profile of the 2nd energy storage in the preclinical test. After the 2nd energy storage is charged over 1.8 V, the stored energy is transferred to a Li-ion battery. Therefore, we successfully demonstrate that the stacked I-TENG can recharge a battery for practical purposes inside the body.”.

Comment 3: Regarding the pacemaker application, this I-TENG could generate ~100 V in vitro output whereas the in vivo output is only a few volts. Given the in vitro power density is $4.9 \mu\text{W}/\text{cm}^3$, the in vivo value should be significantly lower. Nevertheless, a commercial pacemaker usually requires 10-20 μW , which is hardly met by the i-TENG harvesting in vivo energy, raising a concern for its practical implantation. This discrepancy needs to be discussed. Meanwhile, the authors should specify how the device charged up the pacemaker. I assume it is done by the electrodynamic transducer instead of in vivo biomechanical energy.

Response 3: The authors appreciate the reviewer's comment. Because the I-TENG is a fully encapsulated system, it generates similar output performance in a laboratory environment or *in vivo* environment conditions. Output performance of the I-TENG is $38.1 \mu\text{W}$ per movement, which is higher than the power consumption of a commercial pacemaker. As the reviewer mentioned, voltage signals from Figure 4a-c is much smaller than output performance of the I-TENG (Figure 2d), because we measured output voltage signal on $60 \mu\text{F}$ capacitors embedded into the wireless monitoring system. Figure 4a that measured by oscilloscope shows output performance of the I-TENG after connecting to the wireless system. Due to the capacitors, its output voltage is around 4 V, which is a similar performance of *in vivo* experiment results (Figure 4b, 4c). As we described in the manuscript, 1st and 2nd energy storages are successfully charged by the animal experiments.

Modifications to manuscript: On page 9, line 206-208, we modified "Output voltage of the five-stacked I-TENG with a low-pass filter, capacitors ($60 \mu\text{F}$ capacity), and the system load was measured by the oscilloscope (Fig. 4a) and the wireless measurement system (Fig. 4b,c).".

Comment 4: The authors noticed the synchronization of different units' movement is of great importance for the output, considering the phases of electricity. They claimed that the design enables mechanical energy applied simultaneously. How did the design affect the mechanical energy input? It is not clear and should be elaborated more.

Response 4: Thank you for the valuable comment. The suspension structure requires a specific spring constant and mass design to facilitate resonant frequency control. This design is suitable for regularly vibrated conditions to amplify the mechanical movements (*Sci. Rep.* **2019**, 9, 8223). On the other hand, the structure proposed in the manuscript is very simple and intuitive. Although stacked I-TENGs are independently operated by an external force, they share the exact same external force. Due to the same force and waveform of the force applied to the five-stacked I-TENGs, the freestanding masses of each I-TENG behave almost identical and each I-TENG generates the same electrical phase (see Figure 2d,e). Furthermore, the proposed design requires only z-axis movement to harvest energy, thus it is advantageous to harvest irregular movements such as body motions.

Comment 5: The authors used amine-functionalized poly(vinyl alcohol) (PVA-NH₂) and perfluoroalkoxy (PFA) as triboelectric materials. Why the authors use these two materials and what are the advantages of these two materials compared to other commonly used ones?

Response 5: We appreciate your comments. Typically, PTFE is used as negative triboelectric material, and metals are used as positive triboelectric material. For the implantable device, device has a limitation of volume, and weight conditions. To maximize the output performance under limited conditions, we utilized PFA that more negative triboelectric material than PTFE (*ACS Appl. Mater. Interfaces* **2016**, *8*, 18519–18525) as a negative triboelectric material. As a positive triboelectric material, we used PVA-NH₂, which has the amine-functionalized group. It has high H atoms, which make PVA-NH₂ triboelectrically more positive than PVA (*Energy Environ. Sci.*, **2019**, *12*, 3156). Although metals are well-known positive triboelectric materials, a metal electrode can increase the potential electrical short-circuit problem between top/bottom metal electrodes and the freestanding Cu electrode. Furthermore, the spin coating process facilitates a micron-thick coating of PVA-NH₂ on the substrate and strong mechanical stability between the substrate and PVA-NH₂. Therefore, we utilized PFA and PVA-NH₂ as triboelectric materials instead of PTFE and metal.

REVIEWERS' COMMENTS

Reviewer #1 (Remarks to the Author):

The authors have addressed all of the comments properly. Thus, I suggest the formal acceptance of this manuscript to be published, and wish the authors with all the best.

Reviewer #2 (Remarks to the Author):

The authors have revised the manuscript carefully according to the reviewer's comments. The quality of this work is greatly improved. The reviewer is satisfied with the revision and has no more other suggestion for this work. I think that this manuscript should be accepted by the journal.

Reviewer #3 (Remarks to the Author):

The authors took the reviewer's comments into serious consideration. They have basically made clear answers to each question point-by-point. Generally, the authors addressed all my concerns and made corresponding changes in the manuscript. This manuscript now is improved, matching the strict requirement of publication. The reviewer, therefore, recommends its publication in Nature Communication without further change.

Reviewer #1 (Remarks to the Author):

The authors have addressed all of the comments properly. Thus, I suggest the formal acceptance of this manuscript to be published, and wish the authors with all the best.

Our response: Authors thank the reviewer for these positive comments and for the reviewer's thorough examination of the text.

Reviewer #2 (Remarks to the Author):

The authors have revised the manuscript carefully according to the reviewer's comments. The quality of this work is greatly improved. The reviewer is satisfied with the revision and has no more other suggestion for this work. I think that this manuscript should be accepted by the journal.

Our response: Authors thank the reviewer for these positive comments and for the reviewer's thorough examination of the text.

Reviewer #3 (Remarks to the Author):

The authors took the reviewer's comments into serious consideration. They have basically made clear answers to each question point-by-point. Generally, the authors addressed all my concerns and made corresponding changes in the manuscript. This manuscript now is improved, matching the strict requirement of publication. The reviewer, therefore, recommends its publication in Nature Communication without further change.

Our response: Authors thank the reviewer for these positive comments and for the reviewer's thorough examination of the text.